# Returning to Nature for the Next Generation of Antimicrobial Therapeutics

**DOI:** 10.3390/antibiotics12081267

**Published:** 2023-08-01

**Authors:** Craig R. MacNair, Caressa N. Tsai, Steven T. Rutherford, Man-Wah Tan

**Affiliations:** 1Department of Infectious Diseases, Genentech Inc., South San Francisco, CA 94080, USA; macnair.craig@gene.com; 2School of Law, University of California, Berkeley, Berkeley, CA 94704, USA; caressatsai@berkeley.edu

**Keywords:** antibiotic resistance, natural product drug discovery, colonization resistance, microbiota-based therapeutics

## Abstract

Antibiotics found in and inspired by nature are life-saving cures for bacterial infections and have enabled modern medicine. However, the rise in resistance necessitates the discovery and development of novel antibiotics and alternative treatment strategies to prevent the return to a pre-antibiotic era. Once again, nature can serve as a source for new therapies in the form of natural product antibiotics and microbiota-based therapies. Screening of soil bacteria, particularly actinomycetes, identified most of the antibiotics used in the clinic today, but the rediscovery of existing molecules prompted a shift away from natural product discovery. Next-generation sequencing technologies and bioinformatics advances have revealed the untapped metabolic potential harbored within the genomes of environmental microbes. In this review, we first highlight current strategies for mining this untapped chemical space, including approaches to activate silent biosynthetic gene clusters and in situ culturing methods. Next, we describe how using live microbes in microbiota-based therapies can simultaneously leverage many of the diverse antimicrobial mechanisms found in nature to treat disease and the impressive efficacy of fecal microbiome transplantation and bacterial consortia on infection. Nature-provided antibiotics are some of the most important drugs in human history, and new technologies and approaches show that nature will continue to offer valuable inspiration for the next generation of antibacterial therapeutics.

## 1. Introduction

In most environments, microorganisms coexist in complex communities and compete with neighboring organisms for space and resources [1]. Microbes have evolved strategies to gain competitive advantages, including nutrient acquisition, metabolic diversity, motility, adhesion, and the secretion of toxic molecules [2]. Utilizing microbial secondary metabolites as life-saving antibiotics has greatly impacted human health and represents just one potential application of compounds produced by these microbes [3,4]. 

Early efforts in antibiotic discovery were centered around the Waksman platform, which involved screening extracts from soil microorganisms, typically Actinomycetes, for antibacterial activity [5]. From the 1940s to the 1960s, this technique ushered in a “golden age” of antibiotic discovery, identifying most of the antibiotics that are still used in the clinic today [6]. Screening for natural product antibiotics fell out of favor as repeated use of extracts from soil microbes led to the rediscovery of known compounds, and in the last 40 years, daptomycin and fidaxomicin are the only two novel classes of natural product antibiotics approved for systemic use in the clinic [7]. In fact, according to current estimates, discovering a single, novel, broad-spectrum antibiotic using the Waksman platform would now require screening over ten million Actinomycetes extracts [8]. This gap in discovery comes at a time when the current antibiotic armament is threatened by the rapid spread of resistance [9].

The need to replenish the pipeline and the futility of screening microbial extracts for novel antibiotics led to a shift towards synthetic-based approaches. However, advances in robotics, increases in the size and complexity of small molecule compound libraries, recombinant DNA technologies, and decades of research investment did not result in a second golden age of antibiotic discovery [10,11]. More often than not, putative leads failed to transition from potent enzyme inhibitors to whole-cell actives. Notably, synthetic chemistry approaches continue to play a crucial role in prolonging the effectiveness of the current antibiotic arsenal by creating analogs with improved spectra of activity and efficacy against resistant pathogens. Unfortunately, the practice of repeatedly modifying the same chemical scaffolds is unsustainable [12]. 

Natural product antibiotic scaffolds have proven to be ideal starting points for antibiotic development and synthetic chemistry programs [13,14]. These scaffolds have been shaped and refined by nature to overcome many of the challenges encountered during antibiotic development, including the ability to penetrate the bacterial envelope, evade resistance mechanisms, and exhibit activity against a wide spectrum of pathogens [15,16]. For example, a key characteristic of β-lactam and glycopeptide antibiotics is their ability to inhibit growth by targeting multiple bacterial processes, a quality that has played a crucial role in the delayed emergence of resistance and sustained clinical effectiveness of these compounds [11,17]. These realizations have sparked a resurgence of interest in identifying novel natural product antibiotic scaffolds. In the first part of this review, we discuss ongoing efforts in this area. Specifically, we describe how new microbial culturing approaches, extraction methodologies, and molecular tools are providing a fresh perspective on antibiotic-producing environmental microbes. We also examine innovative methods that have enabled the identification of novel antimicrobials from “unculturable” microbes through in situ cultivation and culture-independent metagenomic mining.

The use of a single broad-spectrum antibiotic to treat uncomplicated infections has an impressive track record of success. With the continuous discovery of new antibiotic scaffolds, this approach can remain at the forefront of antibacterial treatment for many years to come. However, consistently replenishing the antibiotic pipeline will be challenging as the use of single-agent therapeutics comes with inherent limitations [18]. In contrast to natural environments, where antibiotic-producing organisms employ multiple and often synergistic mechanisms to compete with neighboring microbes [2], the single-agent treatment strategy relies solely on the activity of one antimicrobial agent. This approach imposes significant selective pressure for the development of resistance, potentially explaining the rapid emergence of resistance in clinical settings compared to natural environments. Fortunately, to overcome this and other limitations of current antibiotic treatments, we can further turn to nature to help develop novel antibacterial therapeutic strategies.

High concentrations of diverse microbes must compete for resources in the mammalian gut. Importantly, commensal microbes provide resistance to pathogenic infections through multiple mechanisms, including nutrient and niche competition, stimulation of host immunity, and the secretion of growth-inhibitory molecules [19,20,21]. Recognizing the incredible capabilities of the gut microbiota, researchers have sought to harness it as a multimodal approach and develop microbiota-based therapeutics. These strategies have the potential to effectively treat and prevent infections while avoiding many of the limitations associated with traditional antibiotics, including resistance development and disruption of the host microbiota. In the second part of this review, we explore the limitations of traditional antibiotic approaches and how an improved understanding of microbial communities and naturally occurring colonization resistance can facilitate the development of alternative therapies. These approaches, such as fecal microbiota transplantation, deliver microbes that outcompete and eliminate pathogens, such as *Clostridium difficile*, with incredible efficiency. Overall, nature continues to provide a framework for the latest developments in antibiotic discovery and microbiota-based therapeutics.

## 2. Natural Product Antibiotic Discovery

### 2.1. Reviving Natural Product Antibiotic Discovery in Traditional Antibiotic Producers

Most of the clinically relevant antimicrobial drug classes, such as tetracyclines, macrolides, aminoglycosides, and glycopeptides, were originally isolated from Actinomycetes, particularly from the genus *Streptomyces* [22]. With bioinformatic tools, it is now possible to identify the co-localized genetic elements, called biosynthetic gene clusters (BGCs), that encode for the production of secondary metabolites such as antibiotics [23,24]. Genome mining of *Streptomyces* for BGCs shows that only a small fraction of the encoded natural products have been successfully isolated [25,26]. The discrepancy between the vast genetic potential of *Streptomyces* and the inability to identify novel antimicrobials from these organisms has been attributed to the inactivity of many BGCs under traditional laboratory growth conditions. Additionally, other bioactive molecules can mask the detection of low-level secondary metabolites [27,28]. Researchers are now leveraging new approaches to tap into previously inaccessible and overlooked secondary metabolites of *Streptomyces* for the next generation of antibiotic leads (Figure 1).

Laboratory growth of bacteria cannot match the complex environment found in nature, potentially explaining the lack of BGC expression in vitro. Altering the culture conditions of antibiotic-producing microbes can induce a stress response that stimulates secondary metabolite production. Modifying growth parameters such as temperature [29], pH [30], and osmolarity [31] can increase the yield of known natural products but has found limited success in identifying novel antimicrobials [32]. On the other hand, metal-induced stress can activate a range of silent BGCs and has been used to discover novel secondary metabolites, including antimicrobials [33,34,35]. For example, adding cobalt to a *Penicillium* sp. marine fungus induced the production of novel polyketides with activity against methicillin-resistant *Staphylococcus aureus* (MRSA) and *Pseudomonas aeruginosa* [35]. Another promising approach to activating antimicrobial production in *Streptomyces* is using small molecule inducers [36]. One such molecule, the antibiotic-remodeling compound **2** (ARC2), was identified by its ability to alter the production of pigmented metabolites in *Streptomyces coelicolor* [37]. This molecule has since been shown to activate a diverse range of silent BGCs in over 50 strains of Streptomyces by impairing fatty acid biosynthesis and inducing a cell stress response [38]. 

In the natural environment, microbes exist in complex communities where they constantly interact and compete via secreted molecules and cell-to-cell contact. These interactions can stimulate the production of metabolites [39,40] that are otherwise not observed in standard laboratory monocultures. Therefore, researchers have used co-culturing to better recreate the natural environment and induce BGC expression. For this technique, strains encoding BGCs are grown in communities alongside other microbes, often those from the same environmental origin. Numerous examples of co-culture triggering activation of silent BGCs and promoting antimicrobial production have been reported [41,42,43,44,45]. In one example, the co-culture of a marine fungus with a marine bacterium produced a compound called pestalone that demonstrated potent activity against MRSA (Table 1) [44,46]. The complexity of co-culture can make the mechanistic elucidation of BCG induction difficult. However, strains that promote iron depletion due to siderophore production have been shown to trigger antibiotic production in some neighboring strains [45,47,48], further linking metal stress to metabolite production. Additionally, co-culturing *Streptomyces* with other microbes that produce subinhibitory concentrations of antimicrobial compounds, such as ionophores, has also been found to induce metabolite production in various strains [49,50]. The induction of stress during growth by some but not all stressors appears to be one mechanism that stimulates the expression of silent BGCs in antibiotic producers.

Another major challenge in finding and isolating new natural product antibiotics is that they can be masked by other highly produced bioactive molecules. For instance, in a random sampling of antibiotic-producing Actinomycetes, about 90% of the strains will make streptothricin [8,51]. Recent improvements in extract preparation, including fractionation, mechanism-oriented screening, and nuclear magnetic resonance (NMR)- or mass-spectrometry-based metabolomics-guided techniques, have shown promise in identifying previously overlooked molecules from microbial cultures [52,53,54]. In one study, a large natural product library was fractionated, and both the crude extracts and purified fractions were screened for antimicrobial activity. The importance of fractionation was highlighted by the observation that in 75% of the instances where a fraction showed antimicrobial activity, the corresponding crude extract did not [55]. 

Bioinformatic prediction tools have also been invaluable in identifying strains harboring BGCs of interest and directing extraction protocols [56,57,58]. For example, a new glycopeptide antibiotic, corbomycin, was identified using phylogenetic-directed purification [56,57]. Corbomycin (Table 1) was shown to block peptidoglycan hydrolases in Gram-positive bacteria and reduce the bacterial burden in vivo [56,57]. Furthermore, CRISPR-Cas9 (clustered regulatory interspersed short palindromic repeats (CRISPR)-associated protein 9) technology has been used to inactivate the genes responsible for streptothricin and streptomycin production in Actinomycete strains. Inactivation of these commonly produced compounds improved sensitivity for detecting overlooked molecules and altered metabolite production, leading to the discovery of rare and novel antibacterials, including previously unknown variants of amicetin and thiolactomycin [28].

The soil is just one example of an environment where bacteria must compete for resources. Microbes that produce molecules capable of killing neighboring species have been found in an incredible range of environments, from deep-sea extremophiles [59,60] to the microbiomes of humans and other organisms [61]. A strain of *Photorhabdus khanii* isolated from a nematode was found to produce a molecule, darobactin (Table 1), that exhibits potent antibiotic activity exclusively against Gram-negative bacteria and is efficacious in mouse infection models [61]. This rare Gram-negative spectrum is dictated by the target of darobactin, BamA [61,62]. Found only in the outer membrane of Gram-negative bacteria, BamA has been the focus of numerous antibacterial discovery efforts [61,63,64,65]. Notably, while darobactin does hit a single target, mutations within BamA that confer resistance also reduce virulence during infection. Thus, screening a previously unexplored microbial community revealed a new antibacterial lead molecule targeting a group of bacteria for which a new drug has not entered the clinic in over 50 years. 

By harnessing just a fraction of the vast metabolic potential of *Streptomyces*, the medical community was able to sufficiently outpace the antibiotic resistance threat for nearly 60 years. Now, as the world confronts an escalating wave of resistance, *Streptomyces* and other antibiotic-producing microbes are again positioned to play a pivotal role in providing the next generation of antibiotic leads.

### 2.2. In Situ Cultivation of Previously Unculturable Microbes

In addition to the difficulties in identifying active antibacterial molecules from cultured strains, another major limitation to the discovery of novel natural product secondary metabolites is the inability to culture environmental isolates. Theoretically, if all the ingredients needed for a microbial species to grow are known and provided, in vitro growth should be possible. However, despite an extensive understanding of bacterial metabolism [66] and improved conventional culture conditions [67,68,69], many bacterial species have remained incompatible with laboratory growth. A central driver of this problem is the interdependence among microbial species in their native ecosystems. In certain microbial communities, factors secreted by neighboring organisms are required to support bacterial growth [67,70,71,72].

One approach to growing ‘unculturable’ microbes is to incubate them in an environment that mimics their native ecosystem [73]. In a proof-of-concept study, two novel and previously uncultured bacterial strains were isolated by placing environmental samples inside diffusion chambers followed by incubation in situ [73]. This approach has since been developed by into a high-throughput platform with isolated microchambers called iChips [74]. A similar approach used single-cell encapsulation and flow cytometry to cultivate ‘unculturable’ microbes from seawater, lake sediment, and soil [75]. Many isolates cultivated by in situ approaches were subsequently domesticated, allowing for growth under more conventional culture conditions [74,75,76,77]. 

Newly domesticated microorganisms provide a source of extracts for screening using the traditional Waksman platform (Figure 2). This approach was used to identify teixobactin, a representative of a novel class of antibiotics with broad bactericidal activity against Gram-positive bacteria [78]. Teixobactin (Table 1) was isolated from a previously unculturable β-proteobacterium, *Eleftheria terrae*, and found to target cell wall synthesis and disrupt lipid II containing cytoplasmic membranes, making spontaneous resistance unlikely [79]. Encouragingly, teixobactin was efficacious against MRSA in multiple mouse infection models and showed no resistance liability during standard laboratory testing [78].

Even when screening extracts from previously unculturable bacteria, the possibility of rediscovering known compounds remains. In one instance, this issue was addressed by a species-selective approach, where extracts active against *Mycobacterium tuberculosis* were counter-screened against *S. aureus*. This process yielded lassomycin (Table 1) from *Lentzea kentuckyensis* sp., which has specific bactericidal activity against growing and dormant *M. tuberculosis* [80].

Expanding the supply of microbes for antimicrobial screening from ‘unculturables’ and uncultured isolates is a promising, non-biased approach with the potential for significant discovery of novel antibiotic antibiotics. However, in situ approaches remain limited, leaving many microbes recalcitrant to culturing and domestication. For these, culture-independent approaches offer an alternative route to laboratory growth.

### 2.3. Culture-Independent Mining for Natural Products

Even with the improvements in culturing techniques, many bacteria still remain unculturable or lack BGC expression in vitro, further stymieing discovery efforts [81,82]. Thus, the true metabolic diversity encoded by the genomes of natural microbes remains to be explored. Recent advances in metagenomics and sequencing technologies have been co-opted to facilitate antibiotic discovery within this largely untapped space. 

Early metagenomic studies followed a relatively simple pipeline for generating culture-independent libraries. Environmental DNA (eDNA) was isolated from a sample of interest, cloned into cosmids, and heterologously expressed in model organisms such as *Escherichia coli.* Individual clones were then screened for the production of molecules of interest using phenotypic assays. This technology proved capable of capturing some BGCs and yielded the production of bioactive molecules without the need to culture the producing organism or coax in vitro expression of BGCs [83,84,85,86]. However, metagenomic-based efforts were less successful than may have originally been expected. An inability to capture large BGCs within a single clone and the use of model host organisms ill-equipped to support diverse natural product production limited the chemical matter compatible with heterologous expression. Additionally, only a small fraction of clones within a non-biased metagenomic library will contain BGCs. Secondary metabolite biosynthesis is estimated to comprise less than 2% of many bacterial genomes [87]. This scarcity necessitated large-scale screening of libraries to identify the few clones producing bioactive molecules. Therefore, the field has shifted away from non-biased metagenomic approaches towards more targeted, sequence-based strategies that enrich libraries for BGCs.

Advancements in the understanding of natural product biosynthesis have identified genes common to biosynthetic machinery that can be used as signals for the presence of BGCs. The genomic regions surrounding genes such as polyketide synthase (PKS) and nonribosomal peptide synthetase (NRPS), both of which are commonly associated with BGCs, can be enriched from eDNA samples or previously assembled metagenomic libraries [88,89]. These amplicons, termed natural product sequence tags (NPSTs), can be mined for the presence of known natural product BGCs and novel gene clusters that do not group with previously characterized sequences. Additionally, promoter and pathway refactoring and optimization of heterologous production strains have increased the number and diversity of natural products available to screen for activity [90,91]. 

NPST-guided approaches have been particularly successful in identifying novel natural products [88,89,92,93]. In one instance, phylogenetic analysis of NPSTs generated from unique soil eDNA samples identified novel calcium-dependent antibiotics [92]. This approach led to the discovery of a clade of previously uncharacterized compounds, later named malacidins, that branched away from known calcium-dependent antibiotics such as daptomycin. These molecules, such as malacidin A (Table 1), were heterologously expressed and found to be a unique class of lipopeptides with antimicrobial activity against Gram-positive bacteria. Notably, malacidins will likely avoid cross-resistance as they inhibit bacterial growth by targeting lipid II, a mechanism distinct from daptomycin. These compounds were efficacious in an MRSA skin infection model and did not select for resistance under standard laboratory conditions. Recently, synthetic routes for malacidin have been described, paving the way for the synthesis of structurally diverse analogs to improve potency and pharmacokinetic properties [94,95].

Bioinformatics tools can now accurately predict the structures of many novel natural products based solely on BGC sequences. Chemical synthesis can then produce the compound of interest without needing heterologous expression or microbial culture [96,97,98]. For example, sequence-guided metagenomics was used to discover new compounds targeting menaquinone biosynthesis, a promising antibiotic target involved in electron transport [96]. After identifying a motif common to all known menaquinone-targeting antimicrobials, NPSTs composed of BGCs from over 10,000 genomes were searched for this conserved sequence. BGCs predicted to encode novel menaquinone-targeting inhibitors were identified, and the structures of these molecules were bioinformatically determined. Total chemical synthesis produced six structurally diverse menaquinone-targeting antimicrobials. Two of these compounds, MBA3 and MBA6 (Table 1), had completely novel chemical structures and were efficacious in treating a murine model of MRSA sepsis [96]. 

The potential of culture-independent metagenomic screening is just starting to be realized. Advancements in synthetic synthesis have the potential to alleviate many of the common hurdles in metagenomic screening, particularly in circumventing the need for successful heterologous expression of large BGCs. Reliance on computational structural predictions and pre-enrichment of BGCs can miss novel natural products whose production is not dependent on NRPS, for example, the previously described ribosomally encoded natural product lassomycin [80]. However, even these have proven tractable to novel discovery. The recently discovered BamA inhibitor darobactin [61] is a ribosomally synthesized and post-translationally modified peptide (RiPP), and a bioinformatic search using genes encoding specific tailoring enzymes identified a novel natural product, dynobactin A, that also inhibits BamA and selectively kills Gram-negative bacteria [99]. Thus, metagenomic-based approaches represent an important strategy that bypasses the difficulties in culturing recalcitrant microbes and providing the ideal conditions to activate BGCs of interest. 

**Table 1 antibiotics-12-01267-t001:** Novel natural product antimicrobials.

Compound	Structure	Target	Discovery Approach	Comment	Ref.
Pestalone	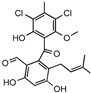	Unknown	Produced by a marine fungus when co-cultured with a marine bacterium.	Potent activity against MRSA and VRE.	[36,40]
Corbomycin	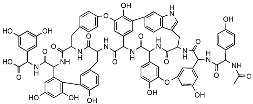	Autolysin inhibition	Phylogenetic analysis of BGCs and resistance determinants predicted production of this novel glycopeptide.	Activity against Gram-positive bacteria.Low levels of resistance development.	[50,51]
Teixobactin	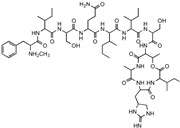	Lipid IILipid III	In situ cultivation of a previously unculturable microbe.	Activity against Gram-positive bacteria.Low frequency of resistance.	[96]
Darobactin	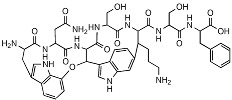	BamA	In situ cultivation of a previously unculturable microbe.	Isolated from a nematode symbiont.Activity against Gram-negative bacteria.	[52,53]
Lassomycin	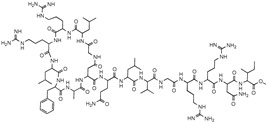	ClpC 1P1P2	In situ cultivation of a previously unculturable microbe.	Narrow spectrum*M. tuberculosis* activity.	[67]
Malacidin A	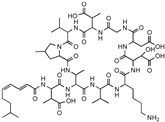	Lipid II	Sequence-guided and culture-independent mining of BGCs.	Structurally distinct calcium-dependent antibiotic.Activity against Gram-positive bacteria.	[82]
MBA6	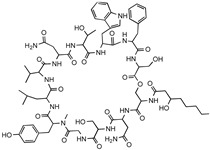	Menaquinone	Sequence-guided and culture-independent mining of BGCs.	Menaquinone-targeting antimicrobials contain a conserved binding motif.Activity against Gram-positive bacteria.	[86]

## 3. Microbiota-Based Therapeutics

Animal and clinical studies suggest that the gut microbiota is important for overall health and protects against the expansion of pathogenic bacteria. Health correlates with high microbial diversity, which limits the growth of unwanted bacteria through mechanisms such as nutrient and niche competition, stimulation of host immunity, and secretion of growth-inhibitory molecules (Figure 3) [19,20,21]. This protection, termed colonization resistance, is a critical function of the healthy microbiota. 

One consequence of traditional antibiotic use is the disruption of the microbiota during treatment, which can cause dysbiosis, a shift in microbial composition and abundance [100,101]. Dysbiosis reduces colonization resistance and increases susceptibility to opportunistic pathogens, including *Enterococcus* spp., members of the Enterobacteriaceae, and *Clostridium difficile* [21,102]. Recognizing the important role of the gut microbiota, researchers have sought to study and harness the mechanisms underlying colonization resistance, aiming to develop microbiota-based therapeutics that treat and prevent infection without many of the limitations of traditional antibiotics.

### 3.1. Colonization Resistance

The role of the gut microbiota in resisting enteric infection was initially discovered in the 1950s when mice treated with the antibiotic streptomycin demonstrated increased susceptibility to infection with *Salmonella* Typhimurium [103]. Since then, researchers have studied the underlying mechanisms behind the colonization resistance provided by a healthy gut microbiota.

Competition for nutrients and niche establishment between commensal and pathogenic bacteria is essential in colonization resistance. Competition in the gut is intense as these microbes have evolved to maximize the utilization of nearly every available substrate and occupy all available niches [104,105,106]. Researchers have sought to exploit the high level of nutrient competition between metabolically related species to reduce the colonization by unwanted bacterial pathogens. For example, commensal *E. coli* strains have been shown to reduce the colonization by pathogenic *E. coli* in mice by exhausting available sugars, amino acids, and micronutrients [107,108,109,110].

Nutrient and metabolite composition changes during gut dysbiosis can also stimulate the growth of and colonization by unwanted bacteria. One molecule that has proven particularly important in regulating *C. difficile* infection are bile acids. Typically, bacteria within the healthy gut microbiota convert primary bile acids into secondary bile acids [111,112]. Antibiotic treatment, however, can eliminate these bacteria from the microbiota, reducing secondary bile acids that are otherwise growth-inhibitory to *C. difficile* [113,114]. Therefore, increasing the availability of secondary bile acids in post-antibiotic patients is an intriguing treatment option to overcome this effect. To this end, *Clostridium scindens* is one of a small subset of commensal species that can convert primary into secondary bile acids, and treatment of *C. difficile*-infected mice with *C. scindens* can reduce *C. difficile* colonization [114]. The direct use of secondary bile acids is also promising, demonstrating in vitro growth inhibitory activity [113,115] and clinical efficacy against *C. difficile* infection in a small number of patients [116].

Colonization resistance also stems from commensal-produced toxins and antimicrobial peptides. Many members of the gut microbiota express type VI secretion systems (T6SS) that deliver growth-inhibitory toxins, called effectors, into neighboring bacteria [117,118]. For example, T6SS in Bacteroidetes has been shown to help establish stable communities of commensal Bacteroidetes while excluding toxigenic strains [119]. The T6SS effectors target various bacterial processes causing cell wall disruption, pore formation, and nucleic acid degradation [120], but their repurposing as therapeutics remains to be explored. Several commensals also produce ribosomally synthesized small antimicrobial proteins or peptides known as bacteriocins or microcins, which inhibit the growth of neighboring bacteria. Studies have shown that producing these structurally and functionally diverse molecules can prevent colonization and displace pathogens from the gut environment [20,121,122,123]. As already noted, the many natural products produced by commensal bacteria represent a relatively untapped source of new antimicrobial molecules. Notably, the increased ability to culture bacteria from the intestinal microbiota has unlocked these microbes for further study [69].

Finally, the gut microbiota also confers colonization resistance by modulating host immunity, secreting metabolites, and directly interacting with host cells. Commensals are critical in the innate immune response. Briefly, they maintain the intestinal epithelium and mucus barrier [124], promote neutrophil recruitment [125], and stimulate Paneth cell antimicrobial peptide secretion [126]. The microbiota also drives adaptive immunity by inducing T-cell differentiation and B-cell activation [127,128]. Thus, the gut microbiota utilizes multiple mechanisms to provide a robust defense against pathogen colonization, all representing potential avenues for therapeutic development.

### 3.2. Fecal Microbiota Transplantation and Bacterial Consortia

Therapeutic strategies that use a single aspect of colonization resistance, such as one bacteriocin, a single secondary bile acid, or one competitive bacterial strain, would fail to capture the breadth and diversity of mechanisms employed by a healthy gut microbiota. Therefore, researchers have looked to reconstitute the microbial composition during dysbiosis by administering the entire gut microbiota from healthy donors using fecal microbiota transplantation (FMT).

Enteric infections that arise from dysbiosis in the gut microbiota, typically after systemic antibiotic use, can be challenging to treat with traditional antibiotics. Given the proximity to antibiotic usage, these infections are often caused by multidrug resistant (MDR) bacteria and are likely to recur in the absence of a restored microbiota. *C. difficile* infections are a paradigmatic example and are strongly associated with antibiotic-induced dysbiosis [129]. In individuals with healthy gut microbiota, *C. difficile* growth is suppressed to low levels and is not a health concern. However, dysbiosis can trigger the proliferation and subsequent release of toxins that cause mucosal inflammation and damage the intestinal epithelium [130]. With the high failure rate of antibiotic treatment, recurrence is seen in 20–35% of patients [131], and fatality rates are commonly reported above 15% [132,133]. Thus, identifying alternative therapeutic options for *C. difficile* is of paramount importance. 

Physicians recognized the importance of a healthy gut microbial community in *C. difficile* infection as early as 1958 when four infected patients were administered FMTs from a healthy donor and rapidly recovered [134]. Despite this early finding and many other reports of the efficacy of FMT against *C. difficile* [135,136,137,138], only in the past decade has widespread acceptance accumulated. Clinical trials have now shown that FMT is superior to conventional antibiotic therapies in treating recurrent *C. difficile* infection, with cure rates of 80–90% [139,140].

FMT works by reconstituting the gut microbiota to resemble the healthy donor and restoring colonization resistance [141]. Dysbiosis is often characterized by an overall loss of bacterial diversity and abundance, particularly within the two most common commensal bacterial phyla, Bacteroidetes and Firmicutes [142]. However, within 24 h of FMT, the composition of the microbial community normalizes to resemble that of the healthy donor, which can last for multiple years [143,144]. Notably, the exact composition of the FMT seems not to impact the efficacy as no donor-specific effects have been identified when treating *C. difficile* [145]. Flexibility in donor selection makes FMT an easily accessible, relatively inexpensive, and highly efficacious treatment option.

Notwithstanding its technical simplicity, FMT leverages the many complex mechanisms underlying colonization resistance in healthy gut microbiota. FMT reduces access to nutritional resources, such as the carbon source sialic acid, and increases the concentration of growth-inhibitory secondary bile acids [141]. Additionally, bacteria found within FMT samples can produce bacteriocins that have antimicrobial activity against *C. difficile* [122,123]. FMT also alters host physiology, signaling for epithelial regeneration, mucin and antimicrobial peptide production, and neutrophil recruitment [141]. The unprecedented multitude of underlying mechanisms contributing to the activity of FMT compared to other antimicrobial approaches is a strength of FMT as it reduces the likelihood of resistance development. Indeed, investigations into patients with recurrent *C. difficile* that failed to respond to FMT did not uncover FMT-resistant *C. difficile.* Failure in these cases was determined to result from systemic antibiotic use post-FMT, improper bowel preparation, or an underlying immune condition such as inflammatory bowel disease (IBD) [146]. Encouragingly, repeated FMT was often sufficient to overcome initial treatment failure in even these situations. 

The success in treating *C. difficile* has attracted interest in using FMT against other bacterial pathogens. The intestinal microbiota can harbor MDR bacteria such as vancomycin-resistant enterococci (VRE) [147], MRSA [148], and extended-spectrum β-lactamase (ESBL)-producing Enterobacteriaceae [149]. This reservoir creates a striking vulnerability in patients undergoing prolonged antibiotic treatment. The associated dysbiosis can lead to a massive expansion in the abundance of these pathogens, progressing into endocarditis, urinary tract infections, and bacteremia. VRE represents a particularly severe healthcare threat as it commonly co-occurs in patients with *C. difficile* infection and has an alarmingly high mortality rate [147,150]. Colonization of VRE in the microbiota has been universally detected in patients who develop bacteremia, making decolonization imperative to reduce transmission and infection. Preliminary studies using FMT to eradicate VRE colonization have demonstrated encouraging success [151,152,153]. In one small-scale study, seven of eight patients carrying VRE were decolonized after a single round of FMT [151]. Promising results have also been observed in patients harboring other MDR bacteria [154,155,156], although some species, such as those within the Enterobacteriaceae, appear more recalcitrant to decolonization with current approaches [157].

The clinical potential of FMT to treat and prevent antibiotic-induced gut infections is immense, however, transferring live microorganisms from healthy donors to sick patients comes with inherent risks [158]. It is now recognized that FMT samples must be pre-screened for antibiotic-resistant pathogens. The use of an unscreened FMT donor sample recently resulted in a patient’s death due to an ESBL-producing *E. coli* [159]. The variability in microbiota composition between donors and the lack of robust characterization of FMT samples, even after screening, pose a significant regulatory hurdle to this therapeutic approach.

Given these concerns, numerous efforts have attempted to identify and develop a predefined assortment of bacterial species capable of replacing FMT. A bacterial consortium made up of pure cultures has the potential to retain the many unique advantages of FMT without the inherent risks associated with the use of raw fecal matter. To this end, many attempts have been made to define a consortium capable of recapitulating the efficacy of FMT against recurrent *C. difficile* infection [114,160,161]. Recently, a consortium of eight commensal Clostridia strains (VE303) [162] completed phase II clinical trials, demonstrating an 80% reduction in the risk of *C. difficile* recurrence eight weeks after treatment [163]. Although promising, a key concern for many consortium-based approaches is that the microbial community in treated patients may lack the long-term stability achieved after FMT.

## 4. Conclusions

One of the main challenges in discovering new antibiotics lies in the scarcity of high-quality antibacterial leads [15,16,164]. Developing analogs within the same chemical framework has become increasingly difficult, leading many pharmaceutical companies to discontinue their antibiotic discovery programs. Government support for identifying novel antibiotic leads has also lagged behind late-stage development initiatives. Programs funded by Carb-X [165] and the New Drugs for Bad Bugs initiative [166] focused on critical late-stage antibiotic development projects, but provided limited support for early discovery. Recent commentaries highlighting the importance of antibiotic lead development [16,167] have encouraged legislators to take aim at filling this discovery void.

Identifying new antibiotic leads is considerably more challenging than screening small-molecule libraries for inhibitors of protein targets, an approach successful in many other therapeutic areas. Studies investigating the mechanisms of action of some of the most successful clinical antibiotics have started to shed light on what makes an ideal antibiotic lead. These efforts have revealed that antibiotic classes such as aminoglycosides, β-lactams, and glycopeptides inhibit bacterial growth through complex and multifaceted mechanisms [168,169,170]. This mechanistic complexity appears to be a key factor in reducing the rate of spontaneous resistance development and prolonging the effectiveness of these drugs [15]. While rationally designing inhibitors with multiple targets is difficult given current capabilities, this multi-pronged approach to growth inhibition has been identified in many natural product antimicrobial compounds, making them a great source for new antibiotic leads. Despite their immense potential, natural product scaffolds often possess a high degree of structural complexity, posing challenges in chemical synthesis, structural modification, and large-scale production.

As creative strategies uncover more novel antimicrobial leads, it is becoming increasingly important to determine which scaffolds should be prioritized for further development. To help identify high-priority leads, various groups have speculated on the characteristics that make for a “perfect” antibiotic [164]. Typically, such a molecule would possess qualities that include evasion of current resistance mechanisms, low susceptibility to spontaneous resistance development, and preferential targeting of pathogenic bacteria to reduce damage to the commensal microbiota [164]. All studies reporting novel antibiotic leads should look to test their molecule for these criteria. Encouragingly, some natural product scaffolds come close to meeting these benchmarks. 

Above, we discussed two examples of promising antibiotic leads, teixobactin and darobactin, identified by Kim Lewis and colleagues. Teixobactin is a potent Gram-positive-active antibiotic that evades current resistance and shows minimal resistance development [78]. However, this molecule lacks Gram-negative activity and indiscriminately kills pathogenic and commensal bacteria. On the other hand, darobactin is a Gram-negative specific antibiotic that is also not susceptible to current resistance mechanisms [61]. Notably, this compound is largely inactive against gut commensals, including Gram-negative symbionts such as *Bacteroides*. Both of these scaffolds represent exciting opportunities for further antibiotic development. 

While the field remains far from introducing an ideal antimicrobial treatment into the clinic for most bacterial infections, encouraging progress has been made toward a superior antimicrobial approach for gut infections. Delivery of live bacteria has shown astonishing results in treating *C. difficile* and decolonizing patients of pathogens such as MDR-enterococci and Enterobacteriaceae. FMT and bacterial consortia deliver billions of bacteria that encode millions of genes and produce thousands of metabolites, providing colonization resistance through a multitude of mechanisms. No resistance has been observed to these treatment strategies, and this approach inherently replaces pathogens with commensal bacteria. Regulatory concerns surrounding the composition of and variability in FMT are a significant hurdle that may be resolved by improvements in metagenomic sequencing and thorough sample characterization. However, for now, well-defined bacterial consortia are a promising alternative. In their current form, consortium-based approaches compromise some of the efficacy and benefits of FMT to reduce toxicity concerns, an unfortunate tradeoff that must be balanced in all antimicrobial approaches. Research efforts should look to further leverage advances in machine learning and artificial intelligence technology to identify bacterial strain combinations and define the appropriate consortia size capable of fully recapitulating the incredible efficacy of FMT. Widespread implementation of this approach will require improvements in manufacturing and quality control to ensure patient safety. 

In this review, we have discussed the limitations, potential and recent advancements of two approaches provided by nature that are poised to contribute to the development of future antimicrobial therapeutics: natural product antibiotic discovery and microbiota-based therapies. However, it is important to acknowledge that other approaches, such as AI-enabled drug discovery, vaccines, combination therapies, and phage, will all play a role in addressing the resistance crisis. Moreover, while this review has focused on microbially produced antimicrobials and approaches, other natural sources, such as molecules from plant species, have also been explored [171,172,173,174]. Medicinal chemistry efforts will also continue to be critical in extending the effectiveness of both current and new antibiotic leads. Long-term solutions to the resistance crisis will require embracing innovative therapeutic approaches that lack the inherent liability of resistance development that has plagued all traditional antibiotic treatment strategies. After years of searching, it has become increasingly clear that the next generation of antimicrobial therapies will come from the same place where antibiotic discovery first began, nature.

## Figures and Tables

**Figure 1 antibiotics-12-01267-f001:**
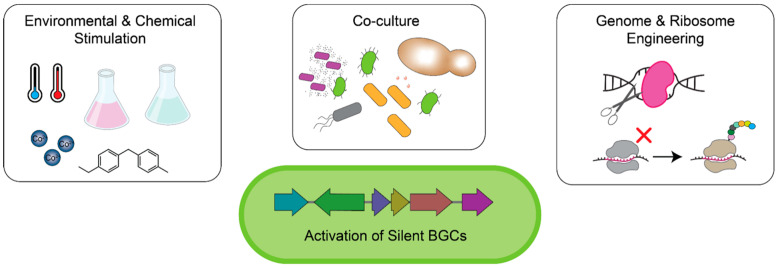
Approaches to elicit production of antimicrobials from silent biosynthetic gene clusters (BGCs). Left: culturing conditions as well as the addition of elicitor molecules such as metals (e.g., Co^2+^) or the antibiotic remodeling compound **2** can alter secondary metabolite production. Middle: co-culturing microorganisms can induce crosstalk through secreted molecules and cell-to-cell contact to activate silent BGCs. Right: genetic modifications to the producing microbe can cause transcriptional rewiring and activation of silent BGCs.

**Figure 2 antibiotics-12-01267-f002:**
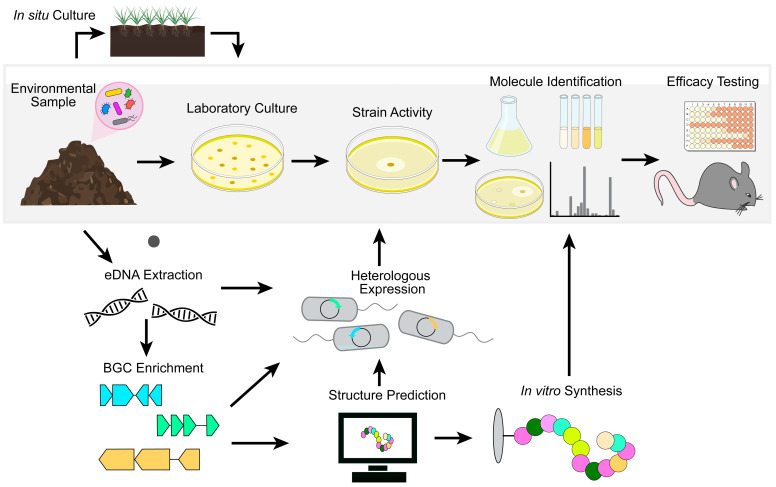
Advances in natural product discovery are revitalizing the Waksman platform. Most approved antibiotics are derivatives of compounds originally discovered with the Waksman platform (gray box). Technological advances are providing new, previously unculturable microbes and chemical matter (e.g., strains heterologously expressing environmental DNA (eDNA), biosynthetic gene clusters (BGCs) enriched from eDNA, or specific BCGs) to screen with the Waksman platform, reviving this approach. Structure prediction also directs the synthesis of novel molecules, which can be structurally confirmed and tested for activity.

**Figure 3 antibiotics-12-01267-f003:**
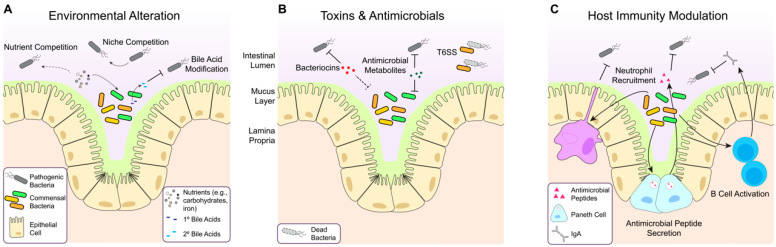
Mechanisms of colonization resistance. A healthy microbiota prevents the expansion of potential pathogens through several mechanisms. (**A**) Commensal microbes can occupy all available intestinal colonization niches. Members of the microbiota also convert primary bile salts (dark blue) into secondary bile salts (light blue) to inhibit pathogens. (**B**) Constituents of the microbiota can express contact-dependent secretion systems that deliver toxigenic molecules to neighboring bacteria as well as antagonistic molecules such as bacteriocins and antimicrobial metabolites. (**C**) The gut microbiota can recruit and stimulate host immunity. Microbial antigens can induce the production of antimicrobial peptides (AMPs) and signal macrophages to promote inflammation and recruit neutrophils. Microbial antigens directly activate B cells, which differentiate into plasma cells and produce protective secretory IgA.

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
