# Peer review of "Returning to Nature for the Next Generation of Antimicrobial Therapeutics"

_antibiotics, 2023, doi:10.3390/antibiotics12081267_

Round 1
Reviewer 1 Report
Overall, the manuscript is well-written and presented with insightful content in the field of antibiotic discovery. To further strengthen the manuscript, several relevant references are suggested to be included. The addition of these references may elevate the quality and impact of the manuscript and provide readers with a broader perspective on the topic.
The author is recommended to consider the following references. For example, references could be added to these paragraphs with no/limited citation at the moment: lines 179 – 181, lines 192 – 196, lines 307 – 310 and lines 552 – 553.
Volynkina, I.A.; Zakalyukina, Y.V.; Alferova, V.A.; Belik, A.R.; Yagoda, D.K.; Nikandrova, A.A.; Buyuklyan, Y.A.; Udalov, A.V.; Golovin, E.V.; Kryakvin, M.A.; et al. Mechanism-Based Approach to New Antibiotic Producers Screening among Actinomycetes in the Course of the Citizen Science Project. Antibiotics 2022, 11, 1198. https://doi.org/10.3390/antibiotics11091198
Zhang, D.; Wang, J.; Qiao, Y.; Lin, B.; Deng, Z.; Kong, L.; You, D. Genome Mining and Metabolic Profiling Reveal Cytotoxic Cyclodipeptides in Streptomyces hygrospinosus var. Beijingensis. Antibiotics 2022, 11, 1463. https://doi.org/10.3390/antibiotics11111463
Hui, M.L.-Y.; Tan, L.T.-H.; Letchumanan, V.; He, Y.-W.; Fang, C.-M.; Chan, K.-G.; Law, J.W.-F.; Lee, L.-H. The Extremophilic Actinobacteria: From Microbes to Medicine. Antibiotics 2021, 10, 682. https://doi.org/10.3390/antibiotics10060682
Cibichakravarthy, B.; Jose, P.A. Biosynthetic Potential of Streptomyces Rationalizes Genome-Based Bioprospecting. Antibiotics 2021, 10, 873. https://doi.org/10.3390/antibiotics10070873
Low, C.X.; Tan, L.T.-H.; Ab Mutalib, N.-S.; Pusparajah, P.; Goh, B.-H.; Chan, K.-G.; Letchumanan, V.; Lee, L.-H. Unveiling the Impact of Antibiotics and Alternative Methods for Animal Husbandry: A Review. Antibiotics 2021, 10, 578. https://doi.org/10.3390/antibiotics10050578
Minor formatting issues found in the manuscript need to be adjusted. E.g. line 146, in-text citation [35–39]; lines 430 – 440, inconsistent font size.
Author Response
Reviewer #1
Overall, the manuscript is well-written and presented with insightful content in the field of antibiotic discovery. To further strengthen the manuscript, several relevant references are suggested to be included. The addition of these references may elevate the quality and impact of the manuscript and provide readers with a broader perspective on the topic.
We thank the reviewer for the overall assessment and suggestions for improvement.
The author is recommended to consider the following references. For example, references could be added to these paragraphs with no/limited citation at the moment: lines 179 – 181, lines 192 – 196, lines 307 – 310 and lines 552 – 553.
Volynkina, I.A.; Zakalyukina, Y.V.; Alferova, V.A.; Belik, A.R.; Yagoda, D.K.; Nikandrova, A.A.; Buyuklyan, Y.A.; Udalov, A.V.; Golovin, E.V.; Kryakvin, M.A.; et al. Mechanism-Based Approach to New Antibiotic Producers Screening among Actinomycetes in the Course of the Citizen Science Project. Antibiotics 2022, 11, 1198. https://doi.org/10.3390/antibiotics11091198
Zhang, D.; Wang, J.; Qiao, Y.; Lin, B.; Deng, Z.; Kong, L.; You, D. Genome Mining and Metabolic Profiling Reveal Cytotoxic Cyclodipeptides in Streptomyces hygrospinosus var. Beijingensis. Antibiotics 2022, 11, 1463. https://doi.org/10.3390/antibiotics11111463
Hui, M.L.-Y.; Tan, L.T.-H.; Letchumanan, V.; He, Y.-W.; Fang, C.-M.; Chan, K.-G.; Law, J.W.-F.; Lee, L.-H. The Extremophilic Actinobacteria: From Microbes to Medicine. Antibiotics 2021, 10, 682. https://doi.org/10.3390/antibiotics10060682
Cibichakravarthy, B.; Jose, P.A. Biosynthetic Potential of Streptomyces Rationalizes Genome-Based Bioprospecting. Antibiotics 2021, 10, 873. https://doi.org/10.3390/antibiotics10070873
Low, C.X.; Tan, L.T.-H.; Ab Mutalib, N.-S.; Pusparajah, P.; Goh, B.-H.; Chan, K.-G.; Letchumanan, V.; Lee, L.-H. Unveiling the Impact of Antibiotics and Alternative Methods for Animal Husbandry: A Review. Antibiotics 2021, 10, 578. https://doi.org/10.3390/antibiotics10050578
We agree that these references can provide additional support for the review and point readers to recent literature that is highly relevant to the topics covered. We have incorporated the following suggested references into the revised manuscript.
Volynkina et al. Antibiotics 2022 – line 162
Zhang et al. Antibiotics 2022 –line 168
Hui et al. Antibiotics 2021 – line 180
Cibichakravarthy et al. Antibiotics 2021 – Line 106
Minor formatting issues found in the manuscript need to be adjusted. E.g. line 146, in-text citation [35–39]; lines 430 – 440, inconsistent font size.
Thank you for the careful reading. We have corrected these formatting issues and carefully reviewed the manuscript for any additional errors.

Reviewer 2 Report
Dear Author;
The manuscript's topic is timely and will be of interest to the journal readers.
Also, the manuscript is very well written, and the ideas flow logically. The review of the literature is thorough, so the reader is given an adequate background about the topic. Also, " Returning to Nature for the Next Generation of Antimicrobial Therapeutics" has been assessed by me. Although it is of interest, we are able to consider it for publication in its current form. I have raised several points which we believe would improve the manuscript and may allow the new version to be published in the Antibiotics.
Pls, see the comments.

Author Response
Reviewer #2
Dear Author;
The manuscript's topic is timely and will be of interest to the journal readers.
Also, the manuscript is very well written, and the ideas flow logically. The review of the literature is thorough, so the reader is given an adequate background about the topic. Also, " Returning to Nature for the Next Generation of Antimicrobial Therapeutics" has been assessed by me. Although it is of interest, we are able to consider it for publication in its current form. I have raised several points which we believe would improve the manuscript and may allow the new version to be published in the Antibiotics.
Pls, see the comments.
We thank the reviewer for the overall assessment and suggestions for improvement.
1: In most environments, microorganisms coexist in complex communities and compete with neighboring organisms for space and resources.. you should have new references?
We have added references for this sentence on line 28.
2: Utilizing microbial secondary metabolites as life-saving antibiotics have greatly impacted human health and represents just one potential application of compounds produced by these microbes. …………Where is references?
We have added references for this sentence on line 32.
3: Natural product antibiotic scaffolds have proven to be ideal starting points for antibiotic development and synthetic chemistry programs. Where is references ? you should add examples too. Such as Ozyigit II, Dogan I, Hocaoglu-Ozyigit A,Kaya Y (2023) Production of secondary metabolites using tissue culture-based biotechnological applications. Front. Plant Sci. 14:1132555. doi: 10.3389/fpls.2023.1132555 2
Though this is not the main topic of this manuscript, we agree with the reviewer that it is worth pointing to such examples. To address this, we have added references to other recently successful examples of this approach (specifically Smith et al. 2018 Nature and Roberts et al. 2022 Nat Comms) at line 56. The suggested reference has been included later in the manuscript (line 548).
4: “Introduction” pls add something more such as what is different between plant base antimicrobial therapeutics and microorganism base antimicrobial therapeutics? And
“There are a number of natural compunds isolated from various sources (plant, animal, or microorganism) that have Antimicrobial Therapeutics. However, due to the structural differences between Gram-negative and Gram-positive bacteria, the efficacy of antimicrobial agents may vary.
This is a useful point from the reviewer. Our review has a specific focus on microbial-based molecules and approaches, but pointing out that there are other natural sources for antibacterials could help the reader. We felt that the Conclusions section was a more appropriate place for this point and have added the following sentences at line 546: “Moreover, while this review has focused on microbial produced molecules and approaches, other natural sources, such as molecules from plant species, have also been explored in recent years.”

Reviewer 3 Report
The current manuscript is an interesting review on the use of nature-derived compounds as a weapon to fight against bacterial infections and antibiotic resistance. It is well presented a reasonably complete, nevertheless some changes should be made before acceptance for publication:
- Subsections should be adequately numbered;
- Figure captions should be summarized; that much text should only exist in the main body;
- Figure 3 should have a better resolution; each individual Figure (A, B and C) is quite small, so I suggest organizing them vertically, instead of horizontally;
- In silico approaches for new drug discovery should be mentioned, as they could be applied to molecules deriving from nature, as well as synthetic ones;
- Semisynthetic approaches (modification of nature-derived molecules) should also be mentioned;
- Since not everything is advantages only, the risks/limitations in the use of natural-derived molecules should be discussed properly.
Author Response
Reviewer #3
The current manuscript is an interesting review on the use of nature-derived compounds as a weapon to fight against bacterial infections and antibiotic resistance. It is well presented a reasonably complete, nevertheless some changes should be made before acceptance for publication:
We thank the reviewer for the comments and suggestions.
- Subsections should be adequately numbered;
The manuscript has now been formatted using the provided Antibiotics layout.
- Figure captions should be summarized; that much text should only exist in the main body;
Figure captions have been shortened.
- Figure 3 should have a better resolution; each individual Figure (A, B and C) is quite small, so I suggest organizing them vertically, instead of horizontally;
A higher resolution version of Figure 3 has been included in the manuscript which should eliminate previous difficulties in reading the figure.
- In silico approaches for new drug discovery should be mentioned, as they could be applied to molecules deriving from nature, as well as synthetic ones;
In silico approaches such as bioinformatics guided biosynthetic gene cluster discovery, structure prediction and synthetic synthesis are highlighted within the manuscript (lines 168-179, 268-317). We have also mentioned AI-enabled drug discovery (line 545) as another encouraging approach to antibiotic discovery.
- Semisynthetic approaches (modification of nature-derived molecules) should also be mentioned;
We agree. We highlight the importance of the semi-synthetic approach in the section starting on line 54 and have augmented this paragraph with some additional references based on this comment and a comment raised by reviewer #2.
- Since not everything is advantages only, the risks/limitations in the use of natural-derived molecules should be discussed properly.
We believe we highlighted many of the limitations that have held natural product antibiotic discovery back throughout the manuscript, but agree with the reviewer that having a specific call out to these would be beneficial to the reader. We have added this to the Conclusions section starting around line 501.
